# Segmentation Reconstruction and Prediction of AIS Trajectory Based on Broad Learning System

Baohua He
Navigation College
Dalian Maritime University
Dalian, China
hebaohua@dlou.edu.cn

Yi Zuo
Navigation College
Dalian Maritime University
Dalian, China
zuo@dlmu.edu.cn

Weihong Wang
Collaborative Innovation Center
for Transport Studies
Dalian Maritime University
Dalian, China

Licheng Zhao
Navigation College
Dalian Maritime University
Dalian, China
zhaolichengzx@dlmu.edu.cn

Tieshan Li
School of Automatic Engineering
University of Electronic Science
Chengdu, China
litieshan073@uestc.edu.cn

C. L. Philip Chen
Computer Science College
South China University of
Technology
Guangzhou, China

*Abstract*— **With the widespread adoption of the automatic identification system (AIS), the collected AIS data has become vast volumes, and cause the analysis and processing of vessel trajectories with highly time consuming. Linear models (LMs) are simple and fast to be widely applied in trajectory reconstruction and prediction. However, existing studies generally pursue individual vessel trajectories independently, and the accuracy and stability of LMs are still unsuitable for generalized processing of a large number of trajectories. To address this limitation, we adopt broad learning system (BLS) to establish a prediction model for trajectories. This paper includes three parts of trajectory segmentation, feature extraction and model training. Firstly, K-means clustering is used to segment the trajectories, and divide them into small pieces based on navigation characteristics. Secondly, considering the time series analysis of trajectory data, the segmented trajectory is processed using first-order and second-order differencing to obtain training data. Finally, by training the time series data of different trajectories, a generalized trajectory prediction can be achieved.**

*Keywords—AIS data, Trajectory prediction, Trajectory component, Segmentation, K-means, Broad learning system*

## I. INTRODUCTION

The automatic identification system (AIS) is an important navigation service for information exchange between ships and between ships and shore stations, and has been widely adopted. The extensive collection of vessel navigation data via AIS enables effective prediction of future vessel trajectories, which is crucial for ensuring navigation safety and improving traffic efficiency [1-2]. However, the large volume of data collected by AIS, particularly dynamic information such as longitude, latitude, speed, and course, which need to be continuously recorded and stored, and present challenges for subsequent data processing and practical application. In particular, trajectory prediction based on AIS data requires a method that can quickly, efficiently, and accurately handle trajectory prediction [3-4].

Several existing methods for trajectory prediction mostly include traditional regression analysis and classic machine learning. Among the traditional trajectory prediction approaches, the primary approaches are linear regression analysis (LRA) and least squares method (LSM) [5-6]. In terms of machine learning-based models, the main techniques include support vector machine (SVM) and artificial neural network (ANN) for trajectory prediction [7-8]. In [5], this study presents a framework for vessel trajectory forecasting based on AIS data, and includes a spatiotemporal multi-graph fusion network (STMGF-Net)to identify the interaction graphs of various navigation modes. In [6], this study proposes an improved LSM for vessel trajectory prediction, so as to better handle the nonlinear and dynamic vessel movements, and enhance the accuracy of trajectory predictions based on AIS data. In [7], this study explores vessel trajectory prediction in the Northern South China Sea, and uses SVM to recognize navigation patterns in vessel trajectories, which provides high ability to outperform traditional linear models in complex maritime environments. In the study of [8], Tang et. al. focus on vessel trajectory prediction based on recurrent neural network (RNN), where the RNN can efficiently handle sequential data of vessel movements and also show its superior performance in trajectory prediction tasks than other models.

This paper applies the broad learning system (BLS) [9] to the prediction of vessel trajectories, and the research purpose of this paper is mainly divided into three parts. Firstly, the K-means method is used to segment the AIS trajectories. The trajectories are divided into several segments based on the navigation characteristics, with the number of segments being an adjustable parameter. This step aims to reduce the complexity of trajectory data so as to enhance efficiency and accuracy of model training and prediction. Secondly, the first-order difference and the second-order difference of the trajectory points within each segment are calculated by the temporal sequence of AIS data. This step captures the dynamic relationships between trajectory

points, and improves the model sensitivity to temporal data and predictive capabilities. Finally, BLS is used to establish the prediction model for the segmented trajectories. By training the temporal sequence data of different trajectories, the number of segments are optimized to achieve generalized trajectory prediction. In numerical experiments, this paper takes the AIS data of Dalian Port as an example to model and predict the trajectories of ships. The discussion of experimental results show that the BLS has higher accuracy and shorter computation time in trajectory prediction. This proves the effectiveness and superiority of BLS in AIS trajectory prediction, providing new ideas and methods for future maritime traffic management and trajectory prediction.

The structure of this paper is organized as follows. Section 1 is the Introduction to introduces the significance of the research background and purpose. Section 2 is the Related Studies to provide an overview of the current researches related to trajectory prediction issues. Section 3 is the Methodology to explain the theoretical foundation of the algorithm proposed in this paper. Section 4 is the Experiments to present the results and analyses. Section 5 is the Conclusion of this study.

## II. RELATED STUDIES

### A. Trajectroy Prediction based on Linear Method

Trajectory prediction is a critical component in various applications, such as maritime navigation, air traffic control, and autonomous driving. Accurate trajectory prediction can enhance safety, optimize routes, and improve efficiency. Among the numerous methods developed for trajectory prediction, linear methods remain popular due to their simplicity, computational efficiency, and ease of implementation.

Linear regression analysis (LRA) is a statistical method used to model the relationship between a dependent variable and one or more independent variables [10]. In trajectory prediction of [5], LRA can be employed to predict the future positions of a moving object based on its past positions, and the basic form of LRA is represented by the Equation (1).

$$y = a_1 x_1 + a_2 x_2 + \cdots a_n x_n + \varepsilon \qquad (1)$$

where $y$ is the dependent variable (e.g., future position), $(x_1, x_2, \ldots, x_n)$ are the independent variables (e.g., past positions), $(a_1, a_2, \ldots, a_n)$ are the coefficients, and $\varepsilon$ is the error term. In [11], Peterson and Hovem analyzed vessel traffic patterns using AIS data and establishes a trajectory prediction model based on LRA. This research explored the reliability of AIS data and the impact of human errors on data quality, and also discussed the application of LRA in data analysis, particularly for preliminary modeling in simple trajectory prediction. These studies cover various applications of LRA in vessel trajectory prediction from preliminary modeling to serving as benchmark models, and provide a comprehensive limitations of linear regression in AIS trajectory prediction [5-6, 10-11].

Least squares method (LSM) is also a statistical technique, which can obtain a fitting line to the give data set by minimizing the square of differences among observed and predicted values. The LSM is a widely used prediction model in trajectory prediction due to the simplicity, efficiency, and effectiveness in minimizing prediction errors. For given the dataset $(x_1, y_1), (x_2, y_2), \ldots, (x_n, y_n)$, the basic form of LSM is represented by the Equation (2).

$$y_i = b_0 + b_1 x_i \qquad (2)$$

where the coefficients $b_0$ and $b_1$ are calculated by minimizing the Equation (3) [6].

$$\text{Minimize} \sum_{i=1}^{n} (y_i - (b_0 + b_1 x_i))^2 \qquad (3)$$

In most cases, LSM can be introduced as a hybrid approach with other methods such as support vector regression (SVR). In [6], LSM-SVR was used as advanced methodology for predicting ship trajectories using AIS data. The primary purpose was to enhance the accuracy and reliability of ship trajectory predictions, which are crucial for ensuring maritime safety and improving navigational efficiency. LSM-SVR is employed for its robust capabilities in handling nonlinear features from the AIS data due to its efficiency in regression and predictive modeling. By integrating these two methods, the study achieved more precise trajectory predictions. In [13], Liu et al. also proposed LSM-SVR for ship trajectory prediction based on AIS data. The proposed model was augmented with a selection mechanism that optimizes the prediction process by selecting relevant data inputs dynamically, and this hybrid approach was also tailored to improve prediction accuracy and computational efficiency in maritime applications. The purpose of this research is to develop a model that can provide accurate and real-time predictions of ship trajectories, which is essential for enhancing maritime safety and efficiency.

### B. Trajectroy Prediction based on Neural Network

Neural networks (NN) have been extensively studied and applied in various trajectory prediction tasks due to the flexibility and strong performance in capturing complex patterns in data. Several types of neural network models have also been utilized for ship trajectory prediction. In [8], a vessel trajectory prediction model is proposed by using Long Short-Term Memory (LSTM) neural networks leverages the sequence prediction capabilities of LSTM to accurately forecast future vessel positions based on historical AIS data. This approach captures temporal dependencies and dynamic changes in vessel trajectories, improving prediction accuracy over traditional methods. The LSTM model is trained on large AIS datasets, ensuring robustness and reliability in real-world maritime navigation applications, enhancing safety and efficiency.

Recently, attention mechanism based models were also included in trajectory prediction. In [14], this paper presented a novel approach to predicting ship trajectories in nearby port waters using an attention mechanism model. Accurate prediction of ship movements is essential for ensuring maritime safety, optimizing port operations, and managing traffic efficiently. Traditional methods, which often rely on physical models or statistical techniques, may not fully capture the complex interactions and dynamic environment of port waters. In [15], this study explored the application of an encoder–

decoder model integrated with an attention mechanism for predicting ship trajectories based on AIS data. The encoder–decoder architecture, commonly used in natural language processing tasks, is well-suited for handling sequential data, making it an effective choice for modeling the movement patterns of ships over time. The model's attention mechanism plays a crucial role in enhancing prediction accuracy by allowing the system to selectively focus on the most relevant segments of the input sequence. This capability is particularly valuable in complex maritime environments, where various factors such as traffic density, navigational patterns, and environmental conditions can influence a ship's trajectory. In [16], the proposed model was based on an encoder-decoder architecture, which processes sequential data from AIS records. The introduction of a dual-attention mechanism allows the model to focus on both temporal and spatial aspects of the ship's trajectory. The first attention mechanism is applied to the temporal sequence of past positions, enabling the model to prioritize certain time steps that are more indicative of future movements. The second attention mechanism focuses on the spatial relationship between the ship and its surrounding environment, taking into account factors such as nearby vessels, navigational constraints, and environmental conditions. This dual-attention approach allows the model to dynamically adjust its focus, leading to more accurate trajectory predictions.

## III. METHODOGICAL DESIGN

### A. Overview

This paper explores the application of BLS for the segmentation, reconstruction, and prediction of AIS trajectories, where AIS data provides critical information for monitoring and predicting the movements of vessels. The main processes of this study is shown in Figure 1 and listed as follows.

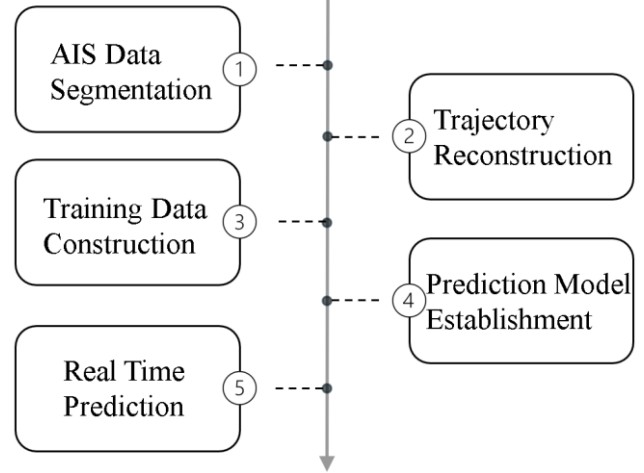

Fig. 1. Overview of the proposed method

(1) AIS Data Segmentation

- Data Collection: AIS data is collected, capturing ship positions, speeds, and timestamps.

- Trajectory Segmentation: The continuous AIS data is segmented into smaller trajectory segments based on specific criteria, such as time intervals or changes in ship direction.

- Feature Extraction: Features like speed, heading, and positional changes are extracted from each segment to characterize the trajectory.

(2) Trajectory Reconstruction

- Reconstruction Algorithm: A reconstruction algorithm is applied to the segmented data to ensure that the trajectory segments are accurately aligned and represent the actual movement patterns of the vessel.

- Interpolation and Smoothing: Missing data points are interpolated, and the trajectory is smoothed to eliminate noise and improve the quality of the data.

(3) Training Data Construction

- Input-Output Pairing: The segments are paired with their corresponding future positions to create training datasets.

- Normalization: Data normalization is performed to ensure consistency across all features, facilitating efficient learning by the BLS model.

(4) Prediction Model Establishment

- Broad Learning System (BLS): The BLS model is established with a flat network structure, which efficiently maps the input features to the predicted ship trajectories.

- Training and Validation: The model is trained and validated using the constructed datasets, with the objective of minimizing prediction error.

- Real Prediction: Once trained, the BLS model is capable of making real-time predictions of ship trajectories based on incoming AIS data.

### B. Trajectory Segmentation Using K-Means

(1) K-means (KM)

KM is a partition-based clustering algorithm that divides a set of data points into K clusters, where each point belongs to the cluster with the nearest mean value. The objective of KM algorithm is to minimize the sum of squared distances between the data points and their respective centroids. The classic KM algorithm follows these steps:

- Initialization: Choose K initial centroids, which can be selected randomly or based on some heuristic.

- Assignment: Assign each data point to the nearest centroid based on the Euclidean distance.

- Update: Recalculate the centroids as the mean of all data points assigned to each cluster.

- Iteration: Repeat the assignment and update steps until convergence, which occurs when the centroids no longer change significantly.

The K-means clustering can be expressed as:

$$\text{argmin}_{\{C_k\}} \sum_{k=1}^{K} \sum_{x_i \in C_k} ||x_i - \mu_k||^2 \qquad (3)$$

where $x_i$ denotes data point, $\mu_k$ denotes the mean value of cluster $C_k$, and $|| \cdot ||^2$ denotes the Euclidean distance.

(2) Trajectory Segmentation Using K-Means

In the context of maritime traffic, ship trajectories are often complex and vary significantly depending on factors such as speed, heading, and environmental conditions. To effectively analyze these trajectories, they can be segmented into distinct phases or patterns using KM clustering. The segmentation process typically involves the following steps (see Figure 2):

- Data Preprocessing: Ship trajectory data $T$ in Eq. (4), usually recorded in the form of time-stamped latitude ($Lat_i$) and longitude ($Lon_i$) coordinates $P_i$ in Eq. (5) are first preprocessed. This process also involves filtering out noise, interpolating missing data points, and normalizing the data.

$$T = \{P_1, P_2, \ldots P_n\} \tag{4}$$

$$P_i = (Lat_i, Lon_i) \tag{5}$$

- Feature Extraction: Key features $F_j$ in Eq. (6) such as latitude, longitude, speed ($Spe_j$) and course ($Cou_j$) are extracted from the trajectory data. These features are used as input for the KM clustering algorithm given in Section B(1).

$$F_j = (Lat_j, Lon_j, Spe_j, Cou_j) \tag{6}$$

- Clustering: The KM algorithm is applied to the extracted features to segment the trajectory into K clusters $C$ in Eqs. (7) and (8). Each cluster represents a distinct phase of the ship's movement, such as cruising, turning, or slowing down.

$$C = \{C_1, C_2, \ldots C_K\} \tag{7}$$

$$C_k = (F_{k,1}, F_{k,2}, \ldots, F_{k,m}) \tag{8}$$

- Analysis: The resulting segments are analyzed to identify patterns, such as common routes, frequent turning points, or areas where vessels tend to slow down.

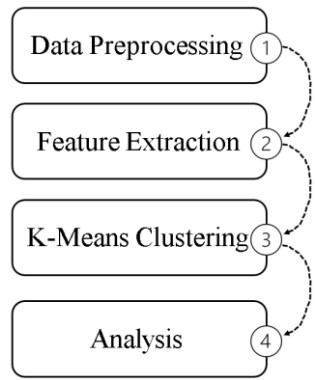

Fig. 2. Flowchart of trajectory segmentation based on KM clustering

C. *Construction of Time-Differential Data*

Time-differential data construction is widely used to extract temporal features from sequential data by calculating the differences between successive data points over time. This approach is particularly useful in time-series analysis, i.e. ship trajectory prediction, where capturing the dynamics of movement is crucial.

Given a time series data of $X$ as shown in Eq. (9), notation $x_t$ denotes the data point at time $t$.

$$X = \{x_1, x_2, \ldots x_T\} \tag{9}$$

The time-differential data can be constructed by computing the difference between successive points as shown in Eq. (10),

$$\Delta x_t = x_t - x_{t-1} \tag{10}$$

where $\Delta x_t$ denotes the time difference of trajectory point at time $t$, $x_t$ denotes the point at time $t$, and $x_{t-1}$ denotes the point at time $t - 1$, respectively. This process transforms the original time series into a new time-differential sequence as shown in Eq. (11),

$$\Delta X = \{\Delta x_2, \Delta x_3, \ldots, \Delta x_T\} \tag{11}$$

which represents the changes in the trajectory over time.

The time-differential data is crucial in contexts where the rate of change is more informative than the original data points. In ship trajectory prediction, the differences in position over time provide insights into the velocity and acceleration of the vessel, which are key factors in predicting future positions.

D. *Trajectory Prediction Using BLS*

(1) Basic concept of BLS

BLS is an emerging learning architecture designed to provide efficient and scalable learning by expanding the network width rather than its depth as shown in Figure 3. This approach is particularly effective for tasks where computational efficiency and rapid model updates are critical.

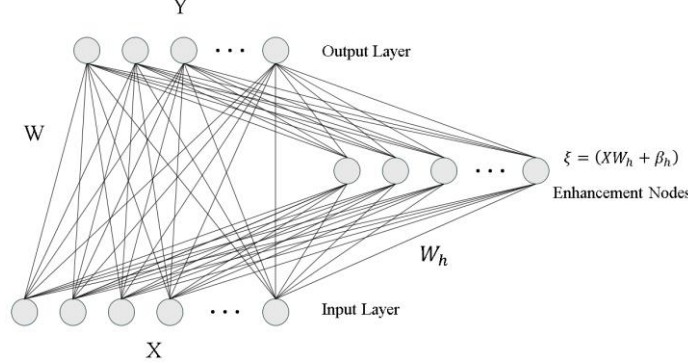

Fig. 3. Flat structure between input and output layer of BLS

BLS was first introduced by C. L. P. Chen and his colleagues as an alternative to deep learning models, which rely on increasing the number of layers to improve performance [17, 18]. Instead, BLS focuses on expanding the width of the network by increasing the number of feature nodes and enhancement nodes as shown in Figure 4. This allows for fast incremental learning and makes the model more adaptable to new data without the need for retraining from scratch.

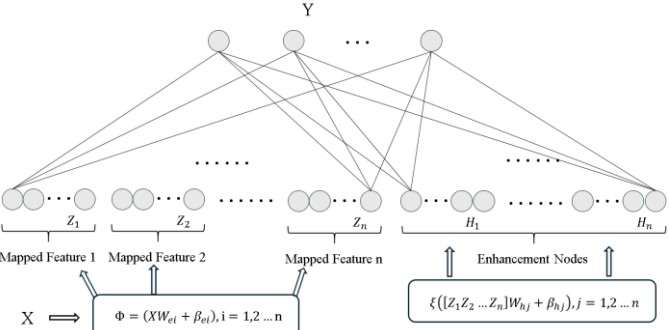

Fig. 4.    Feature and Enhancment layers of BLS

The architecture of BLS includes two main types of nodes: feature nodes and enhancement nodes. The feature nodes are directly connected to the input data, while the enhancement nodes are used to improve the learning capacity of the model. In feature nodes construction, given an input matrix $\mathbf{X} \in \mathbf{R}^{n \times m}$, where $n$ is the length of input data, and m is the size of features. The feature nodes can be given as

$$Z_i = \sigma(XW_i + b_i) \tag{11}$$

where $W_i$ is the weight of feature layer, $b_i$ is the bias term, and $\sigma(\cdot)$ is the active function. In enhancement nodes construction, they are generated from the feature nodes to extract more complex relationships by Eq. (12),

$$H_j = \varepsilon(ZV_j + c_j) \tag{12}$$

where $V_j$ is the weight of enhancement layer, $c_j$ is the bias term, and $\varepsilon(\cdot)$ is also the active function. The final output of BLS is obtained by combining both of the feature and enhancement nodes. The learning process in BLS involves solving a linear system to determine the output weights, which can be done using a pseudoinverse operation as Eq. (13),

$$O = YP^+ \tag{13}$$

where $O$ denotes the weight of output layer, $Y$ denotes the output, and $P^+$ denotes the pseudoinverse of weights concatenated by the outputs of the feature and enhancement nodes as Eq. (14).

$$P^+ = (AA^T + \lambda I)^{-1}A^T \tag{14}$$

When given the trajectory data $(Lat_i, Lon_i, Spe_i, Cou_i)$ as input, the BLS is trained by the sequence at time $t$, and the future trajectory positon at time $t + 1$ can be obtains as shown in Fig.5.

$(Lat_{t-i}, Lon_{t-i}, Spe_{t-i}, Cou_{t-i})$
$\vdots$
$(Lat_{t-2}, Lon_{t-2}, Spe_{t-2}, Cou_{t-2})$ $\Longrightarrow$ BLS $\Longrightarrow$ $(Lat_t, Lon_t)$
$(Lat_{t-1}, Lon_{t-1}, Spe_{t-1}, Cou_{t-1})$

Fig. 5.    Trajectory prediction based on BLS

## IV.  EXPERIMENTS AND ANALYSES

### A.  Data Preparation

The raw AIS data covers global waters and amounts to approximately 72GB, comprising about 1.3 billion AIS records. This vast dataset was imported into database for storage and retrieval. To simplify the data processing, Dalian Port waters were selected as the study area, resulting in the extraction of 4.08 million AIS records. From this subset, 389 AIS records of a specific ship from October 1, 2016, were further selected for detailed analysis.

Given the characteristics of AIS data, clustering techniques were applied to the raw AIS data to enhance the fitting process. The main idea is to cluster the AIS data into different groups and then fit the data within each cluster separately, which helps in capturing the patterns more effectively. The analysis focuses on latitude and longitude data, and the comparison schemes are divided into three categories.

A. Unprocessed raw data: The AIS data is used in its original, unmodified form. This raw data includes the direct measurements from the AIS system, such as timestamp, latitude, longitude, speed, and course. This serves as a baseline for comparison, allowing for the analysis of how processing techniques affect the data fitting and modeling performance.

B. First-order difference data : The data is transformed using a first-order time difference as given in Section III.C, which calculates the difference between consecutive data points to emphasize changes over time.

C. Second-order difference data : The data is transformed by first-order difference data, which is also calculated by the process of Section III.C.

These three data categories (A, B, and C) allow for a comprehensive analysis of the AIS data, with each category providing different insights into the movement patterns of vessels.

### B.  Evaluation Index

The root mean square error (RMSE) is a widely used measure of the differences between predicted values by a model and the actual observed values. It provides a standard deviation of the residuals (prediction errors), which helps in understanding how concentrated the data is around the best-fit line. The RMSE formula is expressed as Eq. (15)

$$\text{RMSE} = \sqrt{\frac{1}{n}\sum_{i=1}^{n}(y_i - y_i')^2} \tag{14}$$

where $n$ is the number of observations, $y_i$ is the actual observed value, and $y_i'$ is the predicted value by the model.

### C.  Prediction Results
(1) Data of Category A

As shown in Figure 6, both of linear model (LM) model and broad learning system (BLS) model exhibit a decreasing trend in RMSE as the number of clusters increases. This indicates that both models improve in accuracy with more refined clustering.

However, the RMSE of the BLS model is consistently lower than that of the LM model across all cluster counts, demonstrating that the BLS model achieves a higher fitting accuracy compared to the LM model.

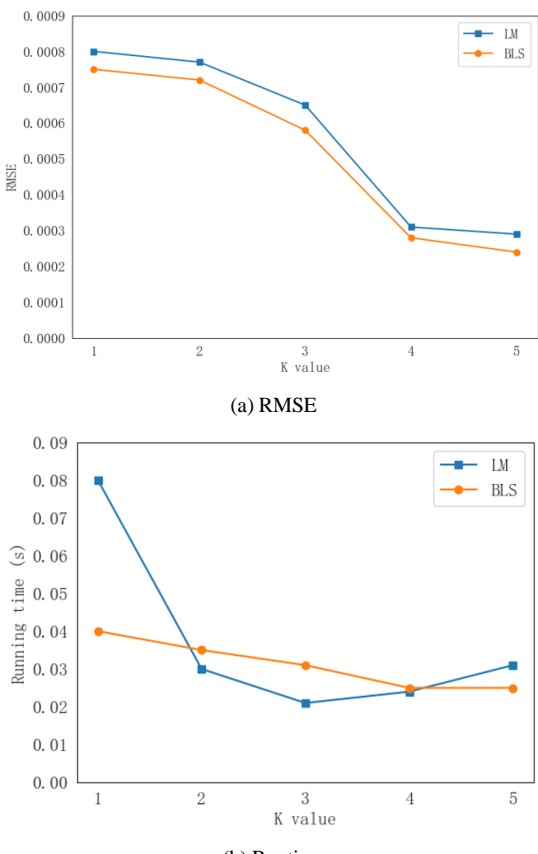

(a) RMSE

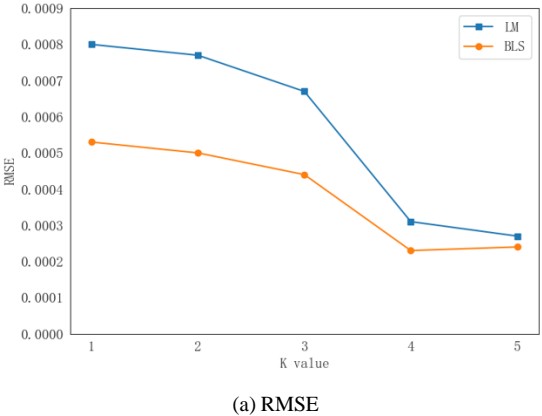

(b) Runtime

Fig. 6. Results comparison in category A

(2) Data of Category B

As shown in Figure 7, both of LM model and BLS model show a consistent downward trend in RMSE as the number of clusters increases. This indicates that increasing the number of clusters improves the fitting accuracy for both models.

However, in every K value, the RMSE of the BLS model is consistently lower than that of the LM model, indicating that the BLS model provides higher fitting accuracy.

(a) RMSE

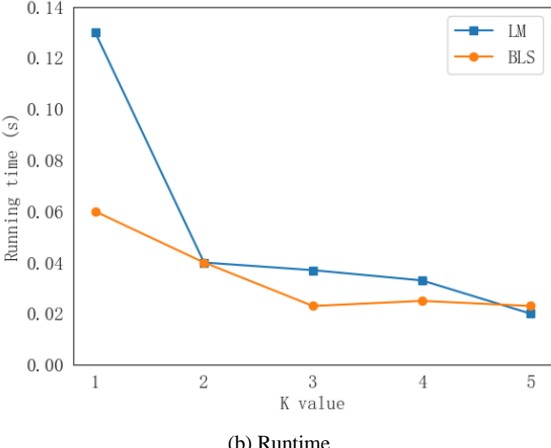

(b) Runtime

Fig. 7. Results comparison in category B

(3) Data of Category C

As shown in Figure 8, both of LM model and BLS model exhibit a consistent downward trend in RMSE as the number of clusters increases, indicating improved fitting accuracy with more clustering.

However, within each K value, the RMSE of the BLS model is consistently lower than that of the LM model, demonstrating that the BLS model achieves higher fitting accuracy.

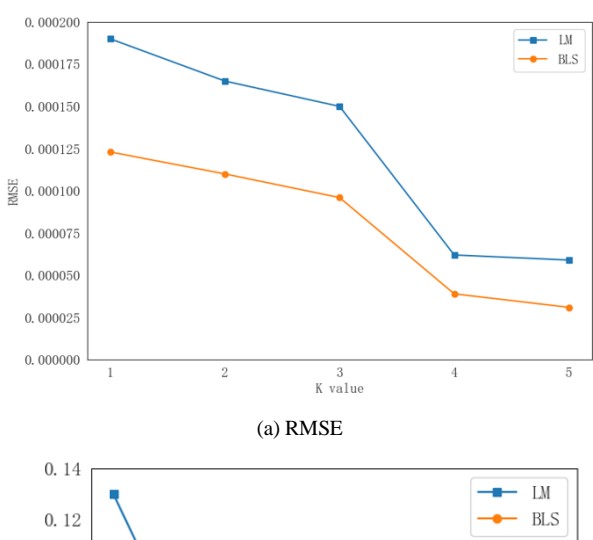

(a) RMSE

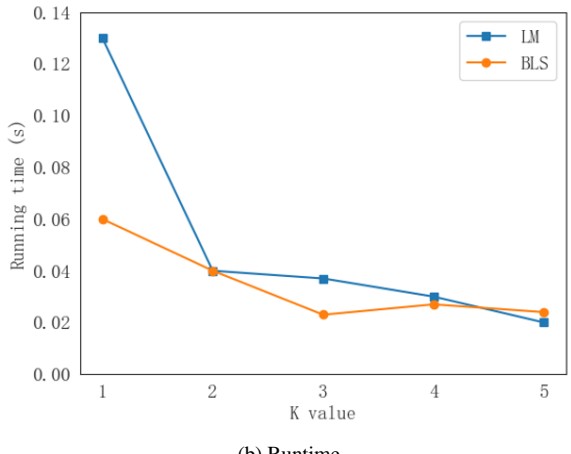

(b) Runtime

Fig. 8. Results comparison in category C

## V. EXPERIMENTS AND ANALYSES

In this paper, we developed a BLS based trajectory prediction model, and applied the KM clustering algorithm to process data within each cluster of trajectory segmentation. The processed LM and BLS models were then used to calculate the case AIS data in Dalian Port. Using latitude and longitude data, three comparison scenarios were established: raw data, first-order difference data, and second-order difference data. The experimental results show that, compared to the segmented LM method, the BLS significantly improves accuracy, exhibits better stability, and has a faster runtime when processing AIS data. The comparative analysis of trajectory prediction methods using BLS provides a foundation for establishing predictive models for large-scale AIS trajectories in the future. It also offers a new predictive approach for maritime authorities to monitor maritime traffic, demonstrating significant exploratory value for practical applications.

## ACKNOWLEDGMENT

This work was supported in part by the National Natural Science Foundation of China (grant nos. 52131101 and 51939001), the Liao Ning Revitalization Talents Program (grant no. XLYC1807046), and the Science and Technology Fund for Distinguished Young Scholars of Dalian (grant no. 2021RJ08).

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
