# OpenReview forum: "Segmentation Reconstruction and Prediction of AIS Trajectory Based on Broad Learning System"
_IEEE.org/ICIST/2024/Conference — IEEE ICIST 2024 Conference Submission_

### Official Review · Reviewer_Xh8i · 2024-08-21
**This article is quite fascinating and of high quality.**

**Rating:** 7
**Confidence:** 3

**Review:**

The paper titled "Segmentation Reconstruction and Prediction of AIS Trajectory Based on Broad Learning System" employs broad learning system to establish a prediction model for trajectories. Firstly, a pseudo-label generator is proposed to label the training samples. First, the K-means clustering method is employed to segment the trajectories, dividing them into smaller segments based on navigation features. Next, considering the time series analysis of the trajectory data, first-order and second-order differencing is applied to the segmented trajectories to obtain the training data. My specific feedback is as follows: 1) What are the practical benefits of generalized processing of a large number of trajectories? The author should further explain the advantages of the article according to the actual situation. 2) Some formatting issues need to be addressed.

---

### Official Review · Reviewer_bV5i · 2024-08-22
**this article presents an interesting and novel fruit shape classification method that effectively combines contour-based and image-based features. The integration of level set and BLS shows promising results, highlighting the potential of such hybrid approaches in agricultural image analysis. However, further improvements are needed to enhance the method's robustness in complex natural environments. The work contributes to the field by proposing an efficient and relatively accurate method for fruit shape classification, which can benefit tasks like automated fruit harvesting and quality inspection. Below is a list of comments that should be taken into account further when revising the paper. 1.	There are a few typos in this paper which should be corrected. And there are some notions missed. Please make some corrections. 2.	Please add the necessary comments for Figures. 3.	Look out for the following grammatical and spelling errors: misspelling，redundant or unnecessary words，subject-predicate consistency problem，punctuation error，preposition error.**

**Rating:** 8
**Confidence:** 3

**Review:**

This paper developed a BLS based trajectory prediction model, and applied the KM clustering algorithm to process data within each cluster of trajectory segmentation. The processed LM and BLS models were then used to calculate the case AIS data in Dalian Port. Using latitude and longitude data, three comparison scenarios were established: raw data, firstorder difference data, and second-order difference data. In general, this work is well organized and appears potentially interesting, it can be accepted with a little modification.
1.	Please explain the derivation of Equation (3).
2.	What is the future direction of the research described in this article?
3.	Please explain what the innovative points of this system are compared to other systems.
4.	Why choose Construction of Time-Differential Data in this paper?

---

### Official Review · Reviewer_pwRn · 2024-08-23
**Segmentation Reconstruction and Prediction of AIS Trajectory Based on Broad Learning System**

**Rating:** 7
**Confidence:** 2

**Review:**

This paper adopt broad learning system (BLS) to establish a prediction model for trajectories. The obtained result is valuable and can be accepted if the following problems can be clarified.
1.	There are still many grammar errors and typos in this paper. For example, an article is missing before BLS in the abstract, a/the should be added, or the plural form should be used.
2.	The abbreviated form should be given at the first occurrence in the article, e.g. AIS, BLS in the introduction. In addition, some abbreviations that are not used multiple times in the article are unnecessary.
3.	In the simulation section, the clarity of the figure is too low, it is recommended to convert the image to '.eps 'format.
4.	References should be formatted consistently.

---

### Decision · Program_Chairs · 2024-09-06

Accept (Oral)